# Syndemic interactions between HIV/AIDS, mental health conditions, and non-communicable diseases in sub-Saharan Africa: A scoping review of contributing factors

Arvin B. Karbasi◉*, Chukwuemeka Iloegbu◉, Christina Ruan◉, Nana Osei-Tutu, Kahini Patel◉, Leah Frerichs, John Patena◉, Dorice Vieira, Deborah Adenikinju, Lydia Samuels, Joyce Gyamfi, Emmanuel Peprah

Department of Global and Environmental Health, Implementing Sustainable Evidence-based interventions through Engagement (ISEE) Lab, NYU School of Global Public Health, New York, New York, United States of America

◉ These authors contributed equally to this work.
* akarbasi18@g.ucla.edu

## Abstract

### Introduction

The syndemic framework provides a critical lens for understanding the complex interplay between HIV/AIDS, mental health (MH) conditions, and non-communicable diseases (NCDs) in Africa. This scoping review explores how these conditions converge to form a syndemic that disproportionately affects vulnerable populations – particularly people living with HIV/AIDS (PLWH). Contextual factors such as stigma, lower socioeconomic resulting in poverty, gender, resource limitations, and fragmented healthcare systems exacerbate these interrelated conditions, posing significant challenges to individuals and their health.

### Methods

A scoping review was conducted to examine the syndemic interactions between HIV/AIDS, MH, and NCDs across Africa. Utilizing the PRISMA-ScR framework and a predefined inclusion criterion, literature searches were conducted in the following databases: PubMed/Medline (OVID), Web of Science (all databases), Web of Science (core collection), Global Health, Cumulative Index of Allied Health Literature (CINAHL), MEDLINE OVID, Psychinfo (OVID), Psychinfo (proquest); and Psychinfo (psychnet) in March 2024. Articles were screened independently by two peer reviewers and conflicts were resolved by a third reviewer. Data were extracted to summarize study characteristics, prevalence rates, and the contextual factors that underpin syndemic interactions among HIV/AIDS, MH and NCDs.

**Data availability statement:** All relevant data are within the paper and its Supporting Information files.

**Funding:** The author(s) received no specific funding for this work.

**Competing interests:** The authors have declared that no competing interests exist.

## Results

An initial search retrieved 5937 articles, with 2913 articles remaining after removal of duplicates. Title and abstract screening further excluded 2706 articles. In total, 207 full-text articles were assessed, of which 17 publications were extracted and included in the review. The scoping review identified a significant prevalence of multi-morbidities amongst PLWH, particularly within hypertension, diabetes, and depression. Women and older adults were disproportionately affected, with gender and age disparities shaping health outcomes. Contextual factors such as stigma, socioeconomic barriers, and fragmented healthcare systems were consistently reported as key contributors to worsening such multi-morbidities. In many publications, NCDs and MH conditions were undiagnosed or poorly managed, complicating HIV treatment and reducing the quality of life. Individual and structural resource limitations, along with poor healthcare integration, further hindered effective care.

## Conclusion

This scoping review underscores the urgent need for integrated healthcare models to address the syndemic of HIV/AIDS, NCDs, and MH in Africa. Interventions should prioritize stigma reduction, capacity building, and comprehensive care to address the underlying socioeconomic determinants of health among PLWH. Strengthening healthcare systems and promoting holistic, patient-centered care is essential for reducing disparities, improving health outcomes, and achieving the Sustainable Development Goals. Future research should expand geographic and demographic coverage to capture the full scope of these syndemic relationships in diverse African contexts.

## 1. Introduction

The syndemic theory provides a distinct lens that emphasizes the role of local socio-cultural, economic, and healthcare infrastructures in shaping the interactions between multiple health conditions [1–5]. Findings from the literature have discussed the interrelatedness and common underpinnings of multiple overlapping disease burdens with social factors in affected individuals, especially in the presence of limited resources, socioeconomic disparities, or weak health systems [6]. Hence syndemics emphasizes the interconnectedness of diseases that interact synergistically within a given geographical location and identifies the contextual factors that contribute to the clustering of these conditions [2] (Fig 1). The syndemic framework allows us to understand and address complex health challenges within populations. In Africa, the convergence of mental health conditions (MH), Human Immunodeficiency Virus (HIV/AIDS), and non-communicable diseases (NCDs) represents a significant and growing public health challenge that remains largely underexplored through a syndemic lens [7,8]. This multifaceted issue is marked by the rising prevalence of NCDs and MH

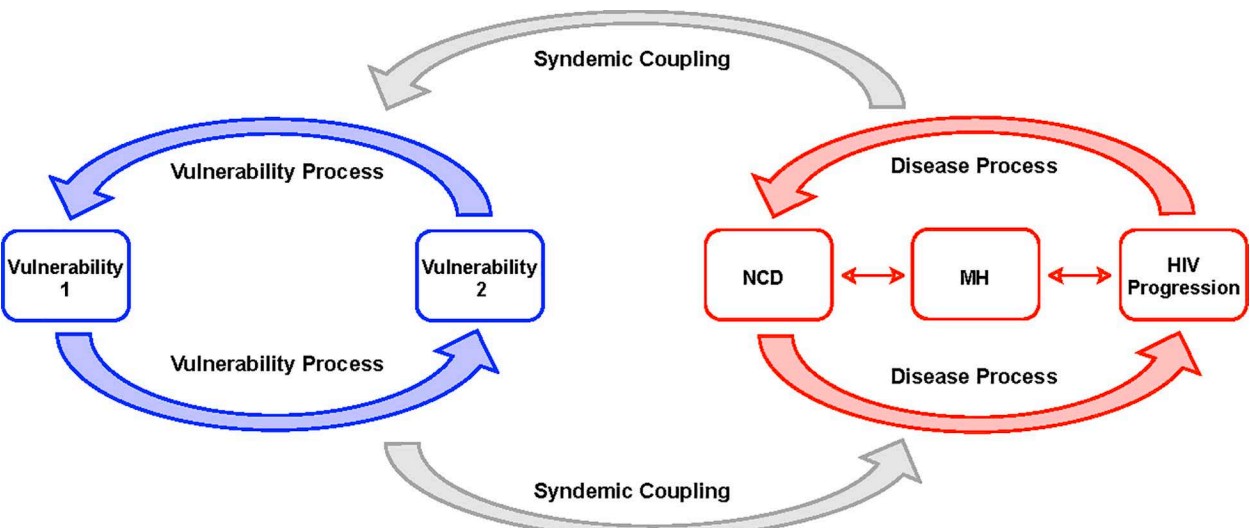

**Fig 1. Syndemic coupling occurs when the progression of diseases such as NCDs, MH disorders, and HIV/AIDS interaction with broader social and structural vulnerabilities, forming a coupled ecosystem that can either exacerbate or mitigate disease outcomes.** Vulnerability processes (represented by the blue arrows), including stigma, poverty, inadequate healthcare infrastructure, interact to reinforce barriers to health and wellbeing. These vulnerabilities can ultimately hinder access to care. The disease process (represented by the red arrows) shows how the progression of HIV, NCDs, and MH disorders can intertwine, intensifying the impact of each condition. Together, the disease and vulnerability process co-occur, creating a dynamic feedback loop that shapes health outcomes through syndemic coupling (shown by the gray arrows).

disorders amongst people living with HIV (PLWH) [9]. The impact of syndemic conditions on fragile healthcare systems is substantial, complicating individual quality of life and straining healthcare resources in low- and middle-income countries (LMICs) [10,11].

The HIV/AIDS epidemic continues to pose a major public health challenge in many African nations, where 25.6 million PLWH and other infectious diseases are on the rise [12,13]. Despite significant progress in the rollout of antiretroviral therapy (ART), stigma, poverty, limited access to healthcare, and challenges with the dissemination of therapies persist as major barriers to improving prevention and treatment efforts [14]. Poverty, for example, is a catalyzing factor for food insecurity which can interact synergistically with pathological co-morbidities to negatively impact an individual's health [6]. Depression is the most common MH disorder affecting up to 30% of PLWH [15]. Collectively referred to as common mental disorders (CMDs), depression and anxiety, are highly prevalent among PLWH, negatively affecting their quality of life and adherence to care [16,17]. In Africa, MH remains a neglected public health issue, with some countries still lacking MH services, according to the World Mental Health Atlas [18–20]. In countries where services exist, the situation remains dire. While the global average of MH professionals across all disciplines is 9 per 100,000, Africa has only 1.4 profession-als per 100,000 people [21]. To reduce such MH challenges, addressing the disparities as it relates to MH professionals in Africa is a point of interest. Similarly, annual mental health outpatient visits in Africa total 94 per 100,000, compared to a global average of 2001 per 100,000 [22]. This situation is only the beginning of a larger issue, while additional challenges exacerbated by high disease prevalence, structural barriers, and individual-level difficulties. Research indicates that the prevalence of MH disorders among PLWH without NCDs is 1.5 to 8 times higher than in the general population world-wide [16,17]. This relationship creates a deleterious cycle, resulting in MH disorders that lead to poorer health outcomes, reduced treatment adherence, and increased comorbidity, further complicating effective responses to the HIV/AIDS epidemic.

Simultaneously, the rising prevalence of NCDs presents new challenges for African nations and poses as a double burden. The World Health Organization (WHO) projects that by 2030, NCDs will account for approximately 44% of all deaths in Africa [23]. NCDs represent the leading cause of death worldwide, killing 41 million people each year—equivalent to 71% of all deaths globally. Among NCDs, the top four causes of significant mortality together account for more than 80% of all premature NCD deaths include cardiovascular diseases (17.9 million deaths annually), cancers (9 million), respiratory diseases (3.9 million), and diabetes (1.6 million) [24]. Moreover, the rising prevalence of NCDs in PLWH adds a dual burden of disease, straining under-resourced healthcare systems within the region, and increases the morbidity and mortality due to the complex interplay between HIV/AIDS, NCDs, and their management [25]. This rise is driven by a combination of factors, including increased life expectancy due to improved ART utilization and adherence, which allows individuals to live longer resulting in age-related health risks [26]. Furthermore, lifestyle modifications and unhealthy habits such as poor diet and physical inactivity, significantly contribute to this trend [27–29]. Finally, limited healthcare access and resources (ex. scant health education, inadequately trained health care and service providers) in many African countries complicate the early detection and management of NCDs, as well as worsening health outcomes for PLWH [30]. NCDs are often linked with MH disorder, as individuals with chronic conditions encounter increased risk of developing CMD including depression due to the psychological burden of managing their health [31,32]. The stress of living with an NCD and HIV/AIDS, combined with societal factors like poverty and limited access for MH treatment creates a bidirectional relationship between these interrelated diseases, and continue to exacerbate each other [33,34] (Fig 1).

The sociocultural context in many African countries further intensifies the syndemic nature of these health issues. Stigmatization, lack of access to healthcare, poor socioeconomic status complicated by low health literacy [35–38] and fragmented care can discourage individuals from seeking necessary treatment. These barriers not only contribute to the persistence of health disparities but also complicate efforts to implement effective evidence-based interventions (EBIs). Addressing these sociocultural factors is essential for fostering environments where individuals feel empowered to seek care and receive the support necessary for their health conditions. However, the interactions among HIV/AIDS, MH disorders, and NCDs extend beyond individual health; they also carry broader public health implications. Addressing these syndemic relationships is crucial for improving health outcomes and achieving the United Nations' Sustainable Development Goals (SDGs), particularly Goal 3, which aims to ensure healthy lives and promote well-being for all [39]. Thus, the development of integrated and evidence-based tailored health interventions that account for the interplay of various co-morbid conditions is essential for tackling complex health challenges. Remedies include increasing awareness as well as screening for MH disorders and NCDs simultaneously while increase capacity to ensure that healthcare systems are equipped to address these interconnected conditions.

Assessing the impact of the relationships between HIV/AIDS, MH disorders, and NCDs in Africa is both timely and critical. While previous reviews have examined the prevalence and incidence of each condition alongside HIV/AIDS [17,26,40–44], there has been no comprehensive scoping review focused solely on the interconnectedness of HIV/AIDS, MH disorders, and NCDs. The purpose of this scoping review is to highlight the hidden syndemic among these three conditions, illuminating the existing literature and providing evidence to inform development and adaption of future EBIs to comprehensively target these complex interactions. To our knowledge, this is the first scoping review to evaluate the MH disorders and NCD amongst PLWH with a synergistic epidemic lens.

## 2. Methods

A comprehensive scoping review was conducted to synthesize the existing body of evidence on the syndemic relationship between HIV/AIDS, MH, and NCDs across Africa. This review aimed to deepen the understanding of how MH challenges amongst PLWH exacerbate their vulnerability to developing NCDs, including hypertension, diabetes, cardiovascular diseases, and chronic respiratory diseases. By examining the interplay between these health conditions, the study sought to highlight the complex pathways through which MH disorders contribute to the rising prevalence and severity of NCDs

among PLWH. This review was guided by the Preferred Reporting Items for Systematic Review and Meta-Analysis extension for Scoping Reviews (PRISMA-ScR) checklist to ensure a comprehensive appraisal of eligible articles [45]. The PRISMA-ScR checklist is provided in S2 Appendix.

### 2.1 Inclusion/Exclusion criteria

Publications were included if they met the following criteria: (i) were conducted in a Sub-Saharan African country; (ii) included PLWH, MH, and NCDs of interest (i.e., hypertension, diabetes, cardiovascular diseases, and chronic respiratory diseases); (iii) published in English or had translated versions available for non-English articles; (iv) took place between 2000 and 2023; and (v) addressed individuals with multi-morbidities. No limitations were placed on the study design, article type, or the age and gender of participants included in the studies. Exclusion criteria for studies were: (i) research that excluded PLWH, MH disorders, or NCDs of interest; (ii) did not address multi-morbidities; (iii) focused on the biology and pathophysiology of HIV/AIDS.

### 2.2 Literature search methods

A comprehensive search strategy was developed to identify literature that met the predefined inclusion criteria. All African countries were included. The literature search was conducted on March 2024. The full search Strategy is provided in S1 Appendix.

The literature search was conducted in the following 10 databases: PubMed/Medline (OVID), Web of Science (all databases), Web of Science (core collection), Global Health, Cumulative Index of Allied Health Literature (CINAHL), PsycINFO (OVID), and PsycINFO (ProQuest). All citations were managed using EndNote 21, a bibliographic management program.

### 2.3 Assessment

All citations were downloaded to Covidence, a web-based software platform, for screening. Peer-reviewed articles were examined and assessed separately by two individual reviewers to reduce bias. Each reviewer pair screened the same articles independently to determine if the inclusion criteria were met by assessing the title and abstract. Further evaluation was then conducted by assessing the entire article at length. Random assignment was utilized to determine screening of articles and disagreements were resolved by a third party. After confirming that the preselected articles met the entire criteria, a data extraction template was developed in Covidence to extract the data. Information highlighting the syndemic relationship between MH, HIV/AIDS, and NCDs were recorded and downloaded from the Covidence platform to an Excel sheet. Inconsistencies in data were subjected to the judgment of a third independent reviewer. The final articles were ultimately chosen by consensus. Title and abstract screening, assessment, and full extraction were conducted by AK, EI, CR, NOT, DV, LF, JP, KP. Data extraction concluded on September 28th, 2024.

### 2.4 Extraction

Data were extracted via a mix of qualitative and quantitative synthesis in the following manner: (i) a descriptive table summarizing the included publications, including the study design, objectives, and location; (ii) a table detailing the sociodemographic characteristics of the study participants; and (iii) a report on the contextual factors that facilitate the syndemic relationship, along with public health recommendations. The studies were reviewed to explore the syndemic relationship among PLWH, NCDs, and MH disorders in Africa.

### 2.5 Quality assessment

The quality assessment (QA) was conducted on all included publications using a modified google form by two independent reviewers. Cross-sectional studies were evaluated based on JBI Critical Appraisal Checklist [46]. Qualitative research

were evaluated based on Critical Appraisal Skills Programme (CASP) checklist [47]. Mixed Method studies were evaluated based on the Mixed Methods Appraisal Tool (MMAT) tool [48]. Published reports that did not have a defined critical appraisal tool for their specific design were assessed via the overall quality of the article.

The quality assessment was evaluated in three categories for each appraisal: low risk, high risk, and some concerns. Low risk indicated that the item on the assessment tool was described and/or well accounted for in the study according to the tool's specifications for determination. High risk indicated that the item of bias was not sufficiently described/and or inadequately accounted for in the study. Unclear/not applicable indicated that there was no information provided in the study to determine if the item of bias was addressed. Data was visualized using robvis visualization tool [49]. All studies were then given an overall assessment on a scale ranging from 1 (poor-quality study and/or study indicates a high risk) to 5 (high-quality study and/or study indicates a low risk).

## 3. Results

### 3.1 Study selection

An initial search retrieved 5937 articles, with 2913 articles remaining after removal of duplicates. Screening of title and abstract further excluded 2706 articles. In total, 207 full-text articles were assessed, of which 17 publications were extracted and included in the results [50–66]. The screening, elimination process, and reasoning for excluding articles are outlined in the PRISMA chart (Fig 2).

### 3.2 Study characteristics

Key study characteristics are summarized in Table 1 and locations of where included studies (n = 17) were conducted are visualized in Fig 3. Nearly one-third of the studies were conducted in South Africa (n = 5, 29%), followed by Tanzania (n = 4, 24%), Zimbabwe (n = 2, 11%), Ghana (n = 1, 6%), Uganda (n = 1, 6%), Cameroon (n = 1, 6%), Rwanda (n = 1, 6%), Malawi (n = 1, 6%), and Kenya (n = 1, 6%). Studies were predominately cross-sectional (n = 7, 41%), followed by qualitative (n = 5, 29%), secondary analysis of cross-sectional data (n = 2, 12%), individual-level and dyadic-level analysis (n = 1, 6%), individual-based multi-disease model (n = 1, 6%), and mixed-methods (n = 1, 6%).

### 3.3 Syndemic interplay of HIV, NCDs, and MH

Publications on the prevalence of complex multi-morbidities among PLWH has predominantly been conducted in East African countries of Zimbabwe and Tanzania (n = 9, 53%) followed by Southern Africa (Fig 3). This geographical focus underscores a significant gap in the broader understanding of these interrelated conditions across other regions of Africa (e.g., West and Northern Africa). Overall, several studies (n = 6, 35%) found that participants often had uncontrolled diseases, and nearly all individuals diagnosed with MH disorders were previously undiagnosed [50,51]. Researchers used existing datasets to develop a model forecasting NCD patterns among PLWH in Zimbabwe [50]. A significant number of PLWH currently experience NCDs including hypertension, with either chronic kidney disease (CKD) (34%), depression (20%), diabetes (10%), cancer (9%), asthma (8%), and stroke (3%) (Fig 4A). Although the epidemiological shift towards NCDs in Africa trends to be similar among PLWH and HIV-negative persons, notable differences by HIV status indicate that the major contributing multi-morbidity profiles amongst HIV-negative individuals have two NCDs, while a large proportion of PLWH have three NCDs [50]. These estimates, however, may be inaccurate due to limited comprehensive data.

A separate study in Tanzania explored multi-morbidity and its association with hospitalizations in aging populations, identified a rapidly growing burden of multi-morbidity, significantly being influenced by factors such as urbanization, food insecurity, and socioeconomic variables [51]. In a sample of 2,299 adults (age 40 yrs and older), 17.1% had two health conditions, 6.2% had three, and 2% had more than four [52]. The study also revealed that approximately 40% of PLWH also had NCDs and depression, underscoring the frequent co-occurrence of these conditions [52]. In South Africa,

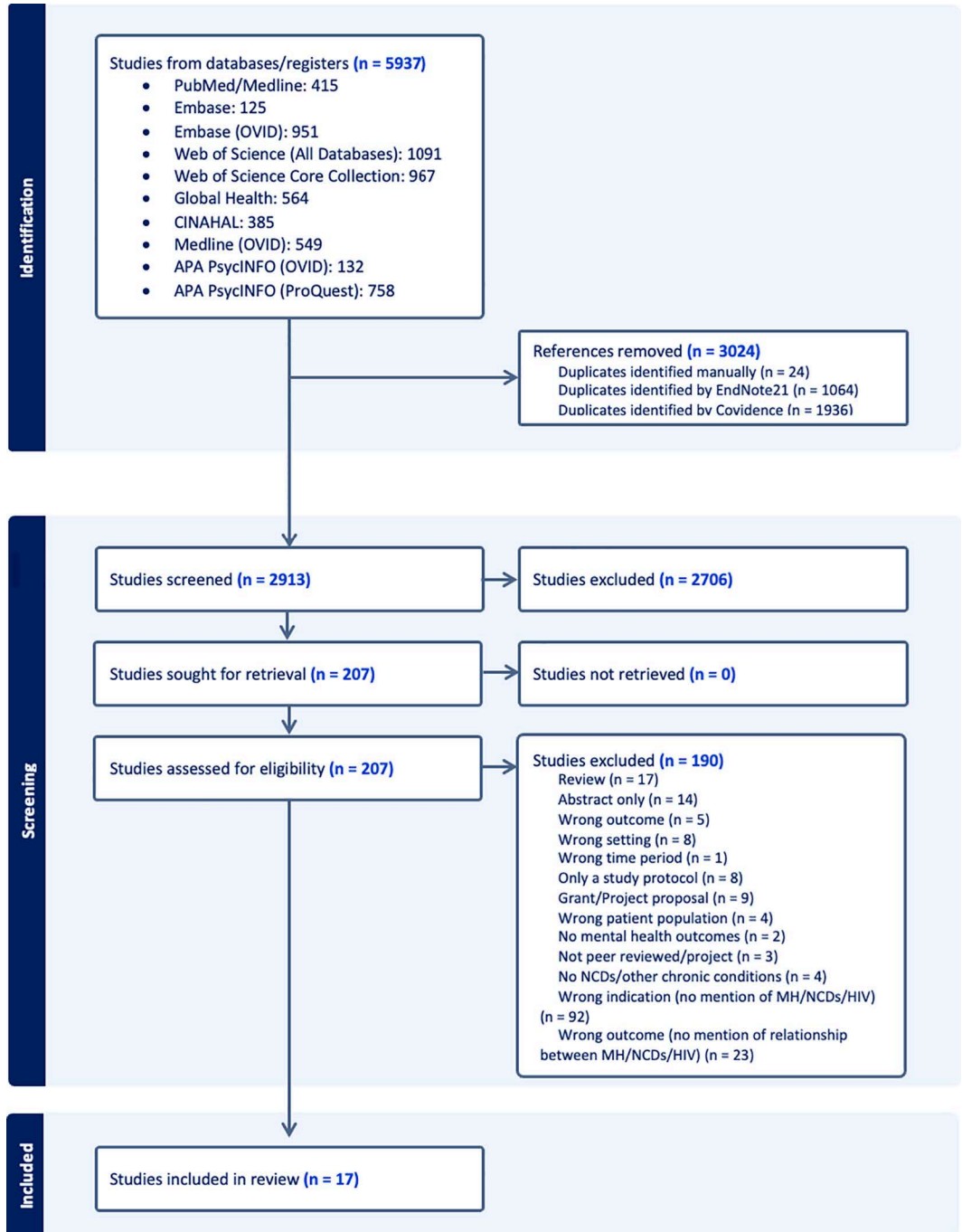

**Fig 2. Preferred Reporting Items for Systematic Reviews and Meta-Analyses extension for Scoping Reviews (PRISMA-ScR) flow diagram.**

participants had an average of 2.3 comorbidities, with 73% of PLWH diagnosed with two or more conditions [53]. Hypertension was the most common comorbidity, among PLWH. While few participants were formally diagnosed with anxiety or depression, all reported high depressive symptoms based on the Center for Epidemiologic Studies Depression Scale (CES-D) suggesting poor diagnosis rates for MH conditions among PLWH [53].

**Table 1. Characteristics of Included Literature.**

| Study ID | Age | Gender (n) | Location | Type of NCD | Type of MH | Contextual Factors |
|---|---|---|---|---|---|---|
| Abdulai, 2022 | 33-85 | Male and Female | Ghana | CVD, DM | Depression | SES<br>FoC<br>Stigma<br>LoR |
| Adedijemi, 2021 | NR | Women (80) and Men (20) | Cameroon | Cancer | Anxiety | Stigma<br>FoC<br>LoR |
| Akugizibwe, 2023 | 25+ | Male (18) and Female (12) | Uganda | CVD, DM | Anxiety | Stigma<br>FoC |
| Bhana, 2017 | 18+ | Male (324) and Female (994) | South Africa | CVD, DM | Depression | FoC |
| Biraguma, 2018 | 18-70 | Female (513) and Male (281) | Rwanda | CVD | Depression | FoC |
| Calderwood, 2024 | 18+ | Female (5215) and Male (1383) | Zimbabwe | CVD, DM, RD | Depression | SES<br>FoC |
| Carpenter, 2022 | NR | NR | South Africa | CVD, DM, RD | General MH | Stigma<br>FoC |
| Chang, 2019 | 40+ | Female (1619) | South Africa | CVD, DM | Depression | Stigma<br>FoC<br>LoR |
| Gooden, 2023 | 20+ | Male (14) and Female (22) | Tanzania | CVD, DM | General MH | SES<br>Stigma<br>FoC<br>LoR |
| Jere, 2023 | 37-66 | Male (25) | Malawi | CVD, DM | Depression | LoR<br>FoC |
| Magafu, 2013 | 18+ | Male (112) | Tanzania | CVD, DM, RD, Cancer | General MH | FoC |
| Mendenhall, 2015 | 35-65 | Male (50) | Kenya | CVD, DM | Depression | LoR<br>FoC |
| Mendenhall, 2019 | NR | Male (30) and Female (50) | South Africa | CVD, DM, Cancer | Depression | SES<br>FoC |
| Mutagonda, 2022 | 18+ | Male (402) | Tanzania | CVD, DM, Cancer | Depression | FoC<br>LoR |
| Qubekile, 2022 | 18+ | Male (37) and Female (63) | South Africa | DM | Depression | FoC<br>SES |
| Smit, 2018 | 19+ | Male and Female | Zimbabwe | CVD, DM, RD, Cancer | Depression | FoC<br>LoR |
| Tomita, 2021 | 40+ | Male and Tomita | Tanzania | CVD, DM | Depression | FoC<br>SES |

Studies are listed in alphabetical order by authors last name; -: No data available; SES; Socioeconomic Status; LoR: Lack of Resources; FoC: Fragmentation of Care; CVD: Cardiovascular Disease; DM: Diabetes Mellitus; MH: Mental Health; RD, Respiratory disease; NR; Not Reported; General MH: No classification of MH.

In Tanzania, a study of 1,318 PLWH found that 14.3% reported common NCDs such as hypertension, diabetes, and cancer, with prevalence ranging from 15% to 58% depending on the setting, likely due to self-reported diagnoses and limited NCD screening [51]. Depression was strongly associated with NCDs, particularly among those with cancer, who were four times more likely to experience depression [51]. Similarly, PLWH diagnosed with an NCD were 3.4 times more likely to score below the MH mean comparison to the general population [54].

A study in Zimbabwe with 6,598 participants found that among individuals over 50, 39% exhibited multi-morbidity, with 7% experiencing complex multi-morbidity involving conditions, such as hypertension, HIV/AIDS, diabetes, and MH

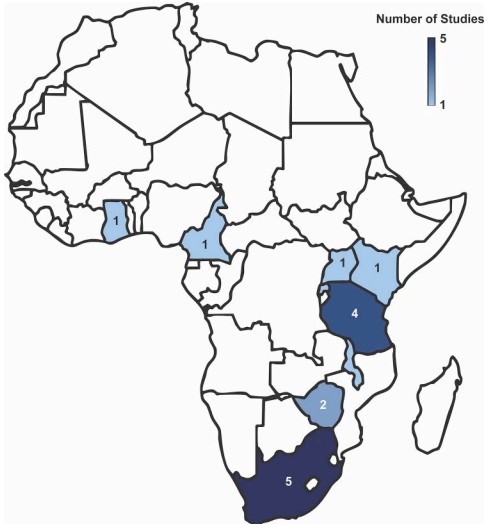

**Fig 3. Heat map displaying the distribution of studies across eight countries, highlighting the relative number of studies in each region.** South Africa has the highest concentration with 5 studies, followed by Tanzania with 4 studies, and Zimbabwe with 2 studies. Kenya, Uganda, Rwanda, Malawi, Cameroon, and Ghana are each represented by 1 study. The color gradient on the map visually reflects the number of studies, with darker shades indicating higher concentrations and lighter shades indicating lower concentrations in each country. Created in BioRender. Karbasi, A. (2025) https://BioRender.com/p7tc1x2.

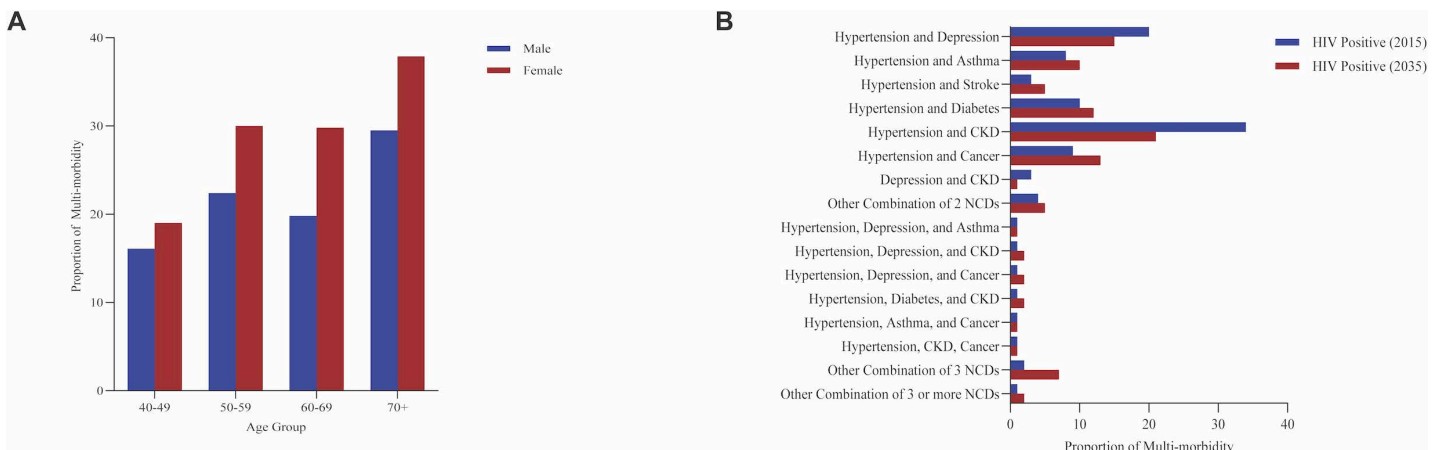

**Fig 4. (A) Prevalence of multi-morbidities across different sex-age groups.** The sample sizes for females and males are as follows: 40-49 years ($n = 1084$), 50-59 years ($n = 607$), 60-69 years ($n = 369$), and 70 + years ($n = 190$). The graph highlights the distribution of multi-morbidity rates by age and sex, providing insights into the variation across different demographic groups. (B) Depiction of the prevalence of PLWH with two or more NCDs in 2015 ($n = 100$) and projected for 2035 ($n = 100$). CKD: Chronic Kidney Disease; NCD: Non-communicable disease.

disorders [55]. Across these five reports, researchers noted that the type of NCD, rather than the number of comorbidities, significantly influenced how PLWH prioritized health conditions. [50–52,54,55]. Patients often prioritized familiar or distressing conditions over those emphasized by clinicians, highlighting the need to improve health literacy among PLWH to better align patient priorities with evidence-based care [53]. These findings highlight the inadequate integration of MH screening into routine clinical care for PLWH, suggesting that the burden of MH challenges is significantly underestimated

in the literature. The results also indicate that MH challenges may play a more substantial role in health outcomes than previously anticipated.

### 3.4 Gender differences and age-related trends in multi-morbidity among PLWH

Two publications conducted in Zimbabwe and Tanzania demonstrated that women are disproportionately affected by multi-morbidity, including hypertension, diabetes mellitus, cervical cancer as well as other significant NCDs [50,52]. One study revealed notable gender disparities in the prevalence of complex multi-morbidities, highlighting that woman aged 60–69 and 70+ experience significantly higher mortality rates compared to men [50] (Fig 4B). Additionally, another study examining the prevalence and predictors of common NCDs among PLWH found that women over 45 years old and weighing more than 75 kg had higher odds of having an NCD, with depression prevalence of 11.8% in this group [51]. Several studies also indicate a higher proportion of women in HIV/AIDS studies, reflecting the greater prevalence of HIV/AIDS among women (6.3% vs. 3.9%) [51]. These observed disparities may be influenced by a combination of methodological and contextual factors, including variations in sampling strategies, differences in research design, and broader societal trends in female participation in research, which may affect both the likelihood of women being included in studies and their willingness or ability to participate.

In South Africa, a study investigating the comorbidity of HIV/AIDS and diabetes mellitus found no significant association with moderate to severe depressive symptoms and clinical variables such as type and complications of diabetes. Gender and educational level were significantly associated with depressive symptoms [56]. These findings are consistent with boarder research showing that women are 1.75 times more likely to experience lifetime depression than men [51,56].

Other reports have underscored the significance of age and HIV/AIDS status in the development of complex multi-morbidities [50–52]. For example, one study revealed that PLWH were more likely to be diagnosed with single or multiple comorbidities compared to HIV-negative individuals. Approximately 33% of PLWH were diagnosed with at least one NCD, compared to 14% of HIV-negative persons [50]. Even without additional NCD risk from HIV or ART, 26% of PLWH would still develop one or more NCDs due to their older average age. This difference in the prevalence of multi-morbidities by HIV status is expected to increase, with projections indicating that the proportion of PLWH diagnosed with at least one NCD will rise from 33% in 2015 to 59% in 2035, and those with two or more NCDs will increase from 5% to 16% over the same period [50].

In two additional publications, the majority of NCDs were either undiagnosed or inadequately controlled, and multi-morbidity was more prevalent among older PLWH [55,57]. Finally, one study exploring the prevalence of diabetes, HIV/AIDS, and MH found that 15% of women and 13% of men, experienced multi-morbidity, with prevalence rates increasing with age [55]. Among individuals aged 50 and older, 39% had multi-morbidity, and 7% had three or more chronic conditions [55]. These findings underscore the growing burden of multi-morbidity among PLWH, particularly among women and older adults, highlighting the urgent need for gender-sensitive and age-appropriate interventions. Addressing these disparities requires targeted screening, early diagnosis, and comprehensive management of NCDs alongside HIV/AIDS care. Additionally, strengthening health systems to improve access to integrated services and addressing the social determinants of health will be essential in mitigating the impact of multi-morbidities on vulnerable populations.

### 3.5 Contextual factors for strengthening holistic care

Several studies emphasized the urgent need to develop and strengthen health systems to improve screening and diagnostics, thereby preventing the progression of these diseases [50–66]. A recurring theme across all reports was the impact of (*i*) addressing fragmented care via integration, (*ii*) resource limitations, (*iii*) socioeconomic determents and (*iv*), HIV-related stigma on health outcomes for PLWH [50–66] (Fig 5). Large-scale success of implementation and scale-up of a 'treat-all' model requires understanding the known barriers to achieving optimal HIV care within a healthcare setting.

MH disorders are common among PLWH, yet they often remain undiagnosed and undertreated in low-resource settings, complicating efforts to develop tailored interventions for syndemic conditions. Organizations in these regions are actively working to address these gaps through policy advocacy, education, and integrating mental health care into existing healthcare systems. In Ghana, ABANTU for development is working to overcome socioeconomic barriers limiting access to NCD treatment for PLWH, despite the availability of free antiretroviral therapy. Through advocacy and educational programs, the organization strives to improve access to comprehensive care [67]. Similarly, in Uganda, StrongMinds is partnering with HIV-focused groups to integrate MH support into HIV/AIDS and NCD care models. Despite facing challenges like limited infrastructure and stigma, their collaboration emphasizes the need for greater investment in a holistic, patient-centered care model [68]. These efforts reinforce the urgency of developing comprehensive training programs that integrate HIV/AIDS, NCD, and MH care to ensure individuals receive the care they need and deserve.

**3.5.1 Integrated care for complex health needs: Addressing the syndemic of HIV, NCDs, and MH in Africa.** Of the several contextual factors identified in the scoping review that influenced the HIV-syndemic, the most notable challenges included fragmented care for individuals with complex multi-morbidities. Fragmented care not only introduces inefficiencies into the healthcare system but is exacerbated as social determinants including poverty, stigma, and inadequate infrastructure, which further complicate these conditions. Studies have found significant associations between fragmented care and negative patient outcomes (e.g., un-necessary testing, increased hospitalization rates and medical costs) [59]. By integrating care and enhancing communication among healthcare providers, systems can address not only individual diseases but also the underlying factors contributing to poor health outcomes [50,51,57,60].

In Uganda, a pilot program integrating HIV, diabetes, and hypertension care demonstrated the benefits of leveraging existing healthcare infrastructure to provide more coordinated care [61]. Integration was also seen as an opportunity to reduce the stigma associated with seeking treatment at HIV clinics, as patients felt that it concealed their HIV status and minimized HIV-related stigma. Participants in the study reiterated that integrating NCDs into HIV care could alleviate stigma while addressing the other barriers that hindered healthcare seeking behavior for comorbid conditions [61]. In South Africa, a country-level analysis further highlighted the growing burden of disabilities; with the combination of chronic HIV/AIDS, rising NCD prevalence, and an aging population necessitates enhanced screening and treatment services. However, these services have not yet been fully implemented [62]. The publication highlighted interventions aimed at

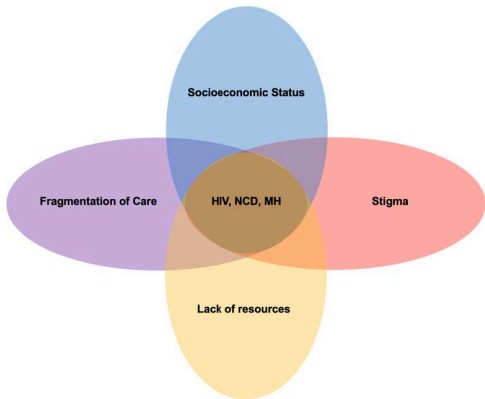

**Fig 5. The interplay between HIV/AIDS, NCDs, and MH syndemics, illustrating how stigma, socioeconomic status, fragmented healthcare systems, and resource scarcity collectively impact health outcomes.** Findings from our scoping review suggest that these interconnected factors, as reported in the literature, significantly influence the co-occurrence of NCDs and MH disorders among PLWH in Africa, exacerbating health inequities and posing substantial challenges to integrated care delivery.

improving identification, linking patients to care, and providing wellness programs could significantly improve public health. Additionally, it stressed the need for integrate MH services into chronic disease management, noting that evidence-based MH interventions are often neglected despite their importance in managing comorbidities [62].

The growing complexity of health needs in South Africa highlights the importance of person-centered, integrated care [53]. While implementing such models in resource-constrained settings is challenging, leveraging local institutional strengths to provide more integrated care is essential for improving patient outcomes [53].

Numerous reports called for enhanced screening and treatment for NCDs and MH disorders within HIV/AIDS care, alongside intensified prevention efforts [50,51,56,57,63]. One study found that care integration could reduce medical costs and improve quality, highlighting one of the major benefits of integrated services [23]. A study in Malawi highlighted that successful integration of care for individuals with complex multi-morbidities depended on strengthening health systems through training and capacity building resulting in improved care quality and better patient outcomes [63]. Additionally, other research emphasized the need for integration via coordinated, patient-centered medical visits, which include comprehensive testing and treatment, as part of a holistic approach to care [64]. Finally, several studies also noted that integration could significantly reduce barriers to access for PLWH, such as discontinuity in care, limited access to healthcare, and the need for greater education on managing these health conditions [51,57,58]. Together, these findings underscore the critical need for integrated, patient-centered care models that address the complex health needs of PLWH, while simultaneously strengthening health systems to improve access, quality, and long-term health outcomes.

**3.5.2 Addressing resources limitations.** Resource constraints can severely hinder the delivery of quality care. The increasing burden of NCDs further strains already fragile healthcare systems, underscoring the need for cost-effective models to manage both HIV and NCDs [50]. Across Africa, research has highlighted the inadequacy of infrastructure for recording and monitoring NCDs and related risk factors, leading to missed opportunities to prevent complex multi-morbidities through essential screening and prevention services [46,51,57,65]. As critical data is not captured and individuals remain unaware of their conditions, there remains a lack of comprehensive understanding and treatment of MH issues and NCDs [50,51,55,57]. Efforts to improve real-time data capture and implement integrated care models have faced significant barriers. For instance, in South Africa examining the implementation of an Integrated Chronic Disease Management model identified staff shortages and drug stock-outs as significant obstacles, leading to fewer patient visits and shorter consultation times with providers [66]. Although some settings have partially integrated care for NCDs, HIV, and MH, full integration remains limited due to ongoing resource constraints at the clinic level [66]. Addressing these challenges requires sustained investment in healthcare infrastructure, workforce capacity, and innovative data management systems to enhance service delivery. Strengthening integrated care models will be essential to effectively manage the dual burden of HIV and NCDs, ensuring that healthcare systems can provide comprehensive, patient-centered care despite resource limitations.

**3.5.3 Socioeconomic determinants of health: The impact of poverty, education, and food insecurity on multi-morbidities in HIV, NCDs, and MH.** A recurring contextual factor is the challenging socioeconomic status of individuals. A study in South Africa found that low educational levels were strongly associated with depression, while higher education had a protective effect that accumulated over the lifetime [50]. Other studies with patients experiencing complex multi-morbidities revealed that many struggled with poverty, often citing difficulties in affording transportation to clinics, medications, and food [11,57,69]. In Ghana, although antiretroviral treatment is provided for free, patients reported challenges in affording medications for hypertension and diabetes, often relying on financial assistance from others [69]. Food insecurity, linked to poor socioeconomic conditions, is also associated with higher rates of multi-morbidities, underscoring the connection between socioeconomic status and health outcomes in individuals living with HIV, NCDs, and MH conditions [69]. These findings underscore the profound influence of socioeconomic factors on health outcomes, particularly for individuals managing HIV, NCDs, and MH conditions. Policies that enhance financial support, expand access to affordable medications (e.g., hypertension and diabetes management) and improve food security can play a

crucial role in mitigating the impact of poverty on health. Furthermore, integrating social and economic interventions into healthcare models may provide a more sustainable approach to managing multi-morbidities in resource-limited settings.

**3.5.4 The role of HIV-related stigma as barriers to care and treatment adherence.** Stigma related to HIV/AIDS, MH, and NCDs emerged as a prominent factor worsening the relationship between these conditions [61,69]. One study found that HIV/AIDS patients with comorbid conditions frequently experienced feelings of depression, loneliness, and hopelessness [69]. HIV-related stigma, coupled with a lack of support, led to worsened emotions – discouraging patients from seeking care or adhering to their treatment for other conditions. Stigma and lack of support were highlighted as significant barriers to treatment adherence and care-seeking behaviors of PLWH [69]. In Uganda, another study observed that the healthcare system was heavily impacted by HIV-related stigma. Participants reported experiencing stigma in various social settings, particularly related to the intersection of HIV and NCDs [61]. Efforts to reduce stigma and promote supportive environments are critical for improving health outcomes among PLWH with comorbid conditions. Implementing community-based interventions, providing mental health support, and fostering patient-provider trust can help mitigate stigma-related barriers. Additionally, integrating stigma reduction strategies into healthcare services may enhance treatment adherence and encourage care-seeking behaviors, ultimately improving overall well-being.

## 3.6 Quality assessment

Quality assessment of qualitative studies (*n* = 5) indicated that at least 80% of all qualitative research studies had a low risk of bias (RoB) across all indicators for the quality of evidence gathered in the studies. RoB assessment of cross-sectional studies (*n* = 7) indicated a high risk of bias amongst measurements of exposures and stated conditions within half of the specific studies. RoB assessment of a singular mixed methods study (*n* = 1) had a low RoB across all indicators for assessing the quality of evidence for this type of study design except for failing to sufficiently address inconsistences between qualitative and quantitative results.

Given the heterogeneity in study designs, a formal risk of bias assessment was conducted only for the qualitative, mixed methods, and cross-sectional studies, as these were the most methodologically appropriate for the appraisal tool utilized and most relevant to the review objectives. Other study types, such as simulation models or secondary analysis, were reviewed for relevance and basic methodological rigor but were not included in the formal RoB scoring (Fig 6).

Final assessment of the included research demonstrated that 29.41% (*n* = 5) received an overall assessment of 5; 41.18% (*n* = 7) received an overall assessment of 4; 29.41% (*n* = 5) received an overall assessment of 3 and none received an overall assessment of 2 or 1. The average assessment was 3.4, indicating that there is a slight risk within the results. One major source of bias that may have affected outcome reporting is that within the cross-sectional designs, self-reporting was utilized. The utilization of self-reporting may have affected the validity of the outcomes by under-reporting or over-reporting the factors discussed at length. Nonetheless, the use of effect measures may have been impossible given the scope of the research question and the nature of the studies themselves. The quality assessment is characterized below (Fig 6).

## 4. Discussion

This scoping review is the first to synthesize current evidence on relationship between HIV/AIDS, MH disorders, and NCDs in the African region. Our review of 17 articles identified several multi-level factors contributing to the co-occurrence of these synergistic epidemics. We found that most of the research were conducted in East and Southern Africa (i.e., Tanzania and South Africa), with a smaller number of reports from other countries within West and Central Africa and no studies in the North African region. Additionally, in one study a correlational matrix was used to discern a relationship between HIV/NCD/MH revealing a particularly strong relationship between HIV and stroke, as well as a notable correlation between stroke and depression, demonstrating that each condition can independently influence the other [52]. This data is consistent with previous publications included in this review understanding complex multi-morbidities within isolated regions in Africa [50–55].

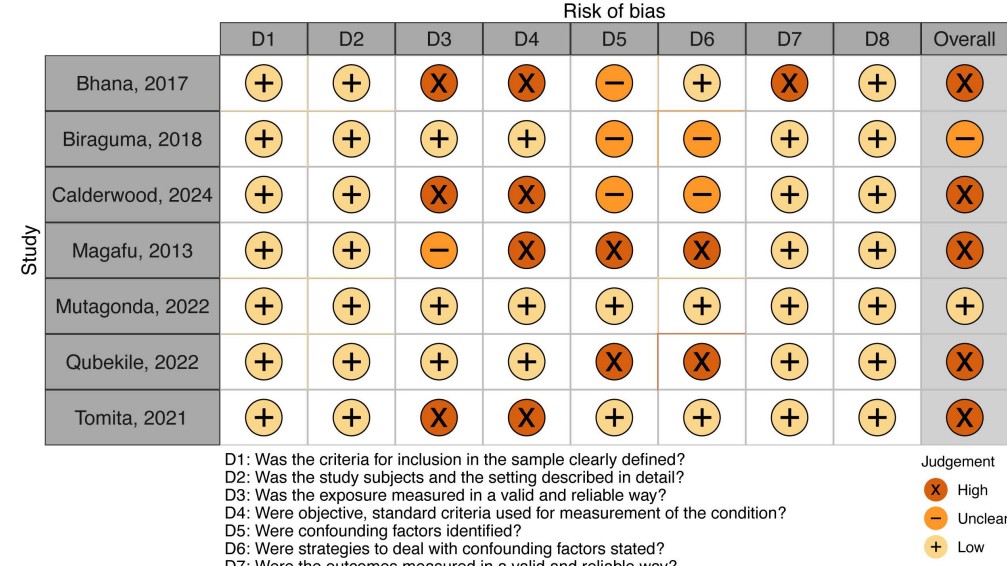

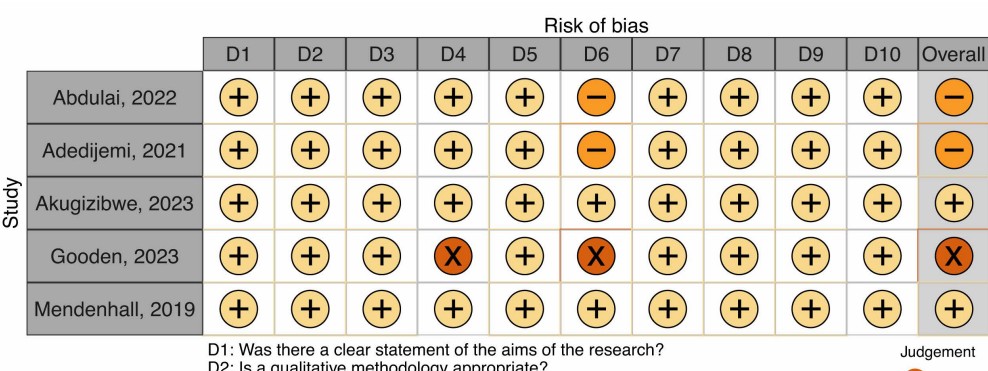

**Fig 6. (A) Quality assessment of qualitative studies (*n* = 5).** Two qualitative studies were found to have an overall low risk, with two have some concerns, and one having high risk. (B) Quality assessment of cross-sectional studies (*n* = 7). One of the cross-sectional studies had low risk, one was found to have some concerns, and five were found to have high risk. (C) Quality assessment of mixed-methods studies (*n* = 1). The lone study included in this paper was found to have high risk.

Contextual factors such as fragmented care, resource scarcity within healthcare systems, stigma, discrimination, and socioeconomic determinants were identified as significant contributors to the exacerbation of these conditions. Disruptions in care, which are often amplified by the co-management of multiple health conditions, can severely impact the HIV/AIDS and NCD care cascade, resulting in reduced care-seeking behaviors and worsened health outcomes for individuals with complex multi-morbidities. Future research should consider collecting data on contextual factors (e.g., care integration, resource limitations, socioeconomic determents, HIV-related stigma) that underpin syndemics occurring to better elucidate these relationships to disease outcomes among PLWH.

The syndemic framework provides valuable insight into the dual burden of MH disorders and NCDs among PLWH. The lack of robust healthcare infrastructure, which fails to accurately capture data from the population, adds to the challenges faced by communities already dealing with longstanding inequities. For instance, a recent study highlighted the lack of data on NCDs and risk factors in the clinical care of adolescents with HIV, underscoring missed opportunities for detecting and addressing comorbidities [70]. Managing the triple burden of HIV/AIDS, MH disorders, and NCDs introduces a new level of complexity and stress for PLWH. Moreover, the difference in the prevalence of multi-morbidities by HIV/AIDS status is expected to increase, with projections indicating that the proportion of PLWH diagnosed with at least one NCD will rise from 33% in 2015 to 59% in 2035, and those with two or more NCDs will increase from 5% to 16% over the same period [50]. While ART has greatly improved life expectancy, with many living into their 60s and beyond, this demographic shift has also increased the prevalence of comorbid conditions, including both physical and MH challenges. Our analysis of various reports support this finding by demonstrating that HIV-associated non-AIDS conditions—such as CVD, diabetes, renal disease, and cancer—are more common in older PLWH [71].

Gender disparities multi-morbidities were also evident, with women living with HIV/AIDS disproportionately affected compared to men. This disparity is shaped by both biological and socio-cultural factors. Women with HIV/AIDS often face unique socioeconomic determinants, such as gender-based violence, caregiving responsibilities, and economic inequality, all of which increase their vulnerability to both physical and MH issues. Women were more likely to report symptoms of depression and anxiety, which, when compounded by HIV-related stressors and multi-morbidities, can negatively affect treatment adherence and overall health outcomes [50,51,56]. Addressing the intersection of gender, HIV/AIDS, and multi-morbidities requires a tailored approach that considers the specific needs of women in HIV/AIDS care models and recognizes both gender and health disparities.

A key finding across literature is the need for improved integration of MH and NCDs screening and services within existing HIV/AIDS care frameworks. Many PLWH also face MH challenges, which exacerbate their overall health outcomes. These intertwined challenges are further complicated by social determinants such as socioeconomic status, HIV-related stigma, and inadequate healthcare resources, all of which hinder access to necessary care. To address these multifaceted issues, a comprehensive and adaptable approach is needed, one that prioritizes integrated service delivery to ensure individuals receive care that addresses their physical, mental, and social well-being. Organizations are addressing gaps in care through community-based interventions, peer counseling, and integrated care models that enhance treatment adherence and access to holistic healthcare. Specifically, in Malawi, resource constraints and fragmented care delay NCD diagnosis and management for PLWH. Partners in Health Malawi strengthens the health system by integrating mental health support into HIV and NCD treatment [72]. In Tanzania, the growing burden of multimorbidity, worsened by food insecurity and economic instability, highlights the need for expanded integrated care. TNW+ provides peer counseling and advocacy for women living with HIV, addressing the intersection of these conditions [73]. In Zimbabwe, Zvandiri's peer-led model helps older PLWH manage depression and cardiovascular diseases, improving adherence and bridging gaps in service integration [74]. These examples highlight the pressing need for scalable, patient-centered solutions that integrate HIV, NCD, and mental health services within existing healthcare frameworks.

Reducing HIV-related stigma and integrating MH is also crucial, as stigma often discourages individuals from seeking care or adhering to treatment. Integrated care models that create supportive, non-judgmental environments can reduce stigma and improve health outcomes by encouraging patients to disclose their health challenges.

Socioeconomic factors and resource limitations play a significant role in amplifying the syndemic between HIV/AIDS, NCDs, and MH. Individuals from lower socioeconomic backgrounds face additional barriers to healthcare access, such as food insecurity, unemployment, and limited education, which hinder effective health management. The reviewed reports highlight how these factors contribute to poor health cycles and mental distress, further reinforcing the syndemic relationship. As the population ages and the prevalence of NCDs rises, there is an increasing need for targeted evidence-based interventions that address the unique challenges posed by resource constraints and socioeconomic disparities [70–74].

### 4.1 Lack of syndemic literature in PLWH in Africa

Our scoping review has some limitations. One of the primary challenges is the limited literature on syndemic interactions among African populations. While syndemics have been thorough examined in high-income countries, there is a dearth of research specifically focused on the unique contextual factors and experiences of PLWH in Africa. The growing burden NCDs among PLWH in the African context highlights the urgent need for more research that takes a syndemics approach. Such research would help identify the distinct socio-economic, cultural, and healthcare system factors that influence the co-occurrence of HIV/AIDS, NCDs, and MH disorders, ultimately improving health outcomes for PLWH in Africa.

Notably, our review also revealed an absence of studies focused on children and adolescents living with HIV. All included studies enrolled adults aged 18 years or older, with many skewed toward older populations. This is an important gap, as children and young PLWH, particularly in relation to mental health and syndemic risk, remain significantly under-represented in the literature. As noted in other studies of HIV-associated MH, youth are often neglected in research on depression and HIV, a pattern seen here [75]. While some NCDs may be more common in older adults, others such as CVD and RD, can have major impacts in younger populations, reinforcing the need to include them in syndemics research [76].

Furthermore, in our scoping review, while we identified common themes across studies, we were unable to assess the methodological rigor of the publications. This lack of detailed assessment limits the ability to determine the quality and reliability of the evidence. The review only included published articles between 2000 and 2023, making it likely to potentially underestimate the burden of the syndemic within the region. Moreover, due to the heterogeneity of the various publications, we were unable to conduct a meta-analysis or aggregate statistical analysis to quantify the prevalence of NCDs and HIV/AIDS comorbidities among PLWH within our scoping review. Therefore, future studies should prioritize inclusion of younger populations, adopt robust and consistent methodologies, and collect comparable data across larger samples to enable more definitive conclusions and inform effective interventions.

## 5. Conclusion

This scoping review summarizes the current evidence on NCDs and MH among PLWH, highlighting the growing prominence of complex multi-morbidities over time. While the publications primarily focused on understanding the relationships between co-morbid conditions and the experiences of individuals living with these conditions, they also revealed significant challenges, including fragmented healthcare systems and a lack of MH literacy among healthcare providers. Furthermore, broader socio-economic factors, such as poverty and stigma, continue to hinder access to quality care. The findings emphasize the urgent need for health system reforms to address these systemic barriers, including improved clinic organization, stronger healthcare infrastructure, and policies that support holistic care for patients with NCDs and MH disorders.

Addressing the HIV/AIDS syndemic of MH, and NCDs requires a comprehensive, integrated approach that not only treats individual conditions but also addresses the underlying social and economic determinants driving these health challenges. Advancing integrated care models, alongside necessary health system reforms and contextually relevant interventions, holds significant potential to improve health outcomes and enhance the overall quality of life for individuals facing the dual burden of MH and NCDs.

## Supporting information

**S1 Appendix. Search strategy.** For our search strategy, the search items included: Non-communicable Diseases (i.e., Cardiovascular Disease, Cancer, Kidney Disease, Respiratory Diseases, and Diabetes), Mental Health (i.e., Mood disorders, substance-use disorders, anxiety disorders, Stigma), and People living with HIV. The search limits included the following: publication year of 2000–2024, no language restrictions, geography of Sub-saharan Africa, and studies focusing specifically on examining the impact of HIV and mental health on the prevalence and management of Non-communicable Diseases.
(DOCX)

**S2 Appendix. PRISMA-ScR-Checklist.** Illustrates the selection procedure for studies including in this scoping review. Additionally, it follows the PRISMA Scoping Review guidelines.
(PDF)

**S3 Appendix. Multi-morbidity trends among people living with HIV.** Demonstrates the prevalence of multi-morbidities across sex and age groups among PLWH (Panel A), and the proportion of individuals with two or more NCDs in 2015 and projected for 2035 (Panel B). Provides a demographic insight into the changing burden of NCDs over time.
(XLSX)

## Author contributions

**Conceptualization:** Arvin Karbasi, Chukwuemeka Iloegbu, Joyce Gyamfi, Emmanuel Peprah.

**Data curation:** Arvin Karbasi, Chukwuemeka Iloegbu, Christina Ruan.

**Formal analysis:** Arvin Karbasi, Chukwuemeka Iloegbu, Christina Ruan, Nana Osei-Tutu, Kahini Patel, Leah Frerichs, John Patena, Dorice Vieira, Deborah Adenikinju, Lydia Samuels, Joyce Gyamfi.

**Investigation:** Arvin Karbasi, Chukwuemeka Iloegbu, Emmanuel Peprah.

**Methodology:** Arvin Karbasi, Chukwuemeka Iloegbu, Nana Osei-Tutu, Dorice Vieira, Emmanuel Peprah.

**Project administration:** Arvin Karbasi, Chukwuemeka Iloegbu.

**Supervision:** Arvin Karbasi, Emmanuel Peprah.

**Validation:** Arvin Karbasi, Chukwuemeka Iloegbu, Emmanuel Peprah.

**Visualization:** Arvin Karbasi, Chukwuemeka Iloegbu, Christina Ruan.

**Writing – original draft:** Arvin Karbasi, Emmanuel Peprah.

**Writing – review & editing:** Chukwuemeka Iloegbu, Christina Ruan, Nana Osei-Tutu, Kahini Patel, Leah Frerichs, John Patena, Dorice Vieira, Deborah Adenikinju, Lydia Samuels, Joyce Gyamfi.

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
