## [Decision Letter · Decision Letter 0]

22 May 2025

PONE-D-25-11960Syndemic Interactions between HIV/AIDS, Mental Health Conditions, and Non-Communicable Diseases in sub-Saharan Africa: A Scoping Review of Contributing FactorsPLOS ONE

Dear Dr. Karbasi,

Thank you for submitting your manuscript to PLOS ONE. After careful consideration, we feel that it has merit but does not fully meet PLOS ONE’s publication criteria as it currently stands. Therefore, we invite you to submit a revised version of the manuscript that addresses the points raised during the review process.

**Overall, a well-written manuscript with a strong rationale and clear methodology. There are a few areas that require clarity and further details:**

**Is there any target population for this review- any specific age, gender etc.? Please clarify. It would be nice to add the names of African countries to the inclusion criteria.****In figure 2, please recheck the calculation- studies screened (2913) - studies excluded (2706) = 207. The total shown in the figure is 188.****Scoping reviews do not typically assess the quality of the studies. Usually systematic reviews appraise studies as one of the core components of their methodology. Figure 6 shows the quality assessment of 13 studies. How about other studies’ findings (for quality)?**

**It would be nice to add a discussion around age distribution in the current evidence. Highlighting any gaps would be useful for future research.**

Please submit your revised manuscript by Jul 06 2025 11:59PM. If you will need more time than this to complete your revisions, please reply to this message or contact the journal office at plosone@plos.org . Please include the following items when submitting your revised manuscript:

We look forward to receiving your revised manuscript.

Kind regards,

Saima Hirani, PhD

Academic Editor

PLOS ONE

**Journal Requirements:**

1. When submitting your revision, we need you to address these additional requirements. Please ensure that your manuscript meets PLOS ONE's style requirements, including those for file naming. The PLOS ONE style templates can be found at https://journals.plos.org/plosone/s/file?id=wjVg/PLOSOne_formatting_sample_main_body.pdf and https://journals.plos.org/plosone/s/file?id=ba62/PLOSOne_formatting_sample_title_authors_affiliations.pdf 2. Please amend either the abstract on the online submission form (via Edit Submission) or the abstract in the manuscript so that they are identical. 3. We note that Figure 3 in your submission contain map images which may be copyrighted. All PLOS content is published under the Creative Commons Attribution License (CC BY 4.0), which means that the manuscript, images, and Supporting Information files will be freely available online, and any third party is permitted to access, download, copy, distribute, and use these materials in any way, even commercially, with proper attribution. For these reasons, we cannot publish previously copyrighted maps or satellite images created using proprietary data, such as Google software (Google Maps, Street View, and Earth). For more information, see our copyright guidelines: http://journals.plos.org/plosone/s/licenses-and-copyright. We require you to either present written permission from the copyright holder to publish these figures specifically under the CC BY 4.0 license, or remove the figures from your submission: a. You may seek permission from the original copyright holder of Figure 3 to publish the content specifically under the CC BY 4.0 license.   We recommend that you contact the original copyright holder with the Content Permission Form (http://journals.plos.org/plosone/s/file?id=7c09/content-permission-form.pdf) and the following text:“I request permission for the open-access journal PLOS ONE to publish XXX under the Creative Commons Attribution License (CCAL) CC BY 4.0 (http://creativecommons.org/licenses/by/4.0/). Please be aware that this license allows unrestricted use and distribution, even commercially, by third parties. Please reply and provide explicit written permission to publish XXX under a CC BY license and complete the attached form.” Please upload the completed Content Permission Form or other proof of granted permissions as an "Other" file with your submission. In the figure caption of the copyrighted figure, please include the following text: “Reprinted from [ref] under a CC BY license, with permission from [name of publisher], original copyright [original copyright year].” b. If you are unable to obtain permission from the original copyright holder to publish these figures under the CC BY 4.0 license or if the copyright holder’s requirements are incompatible with the CC BY 4.0 license, please either i) remove the figure or ii) supply a replacement figure that complies with the CC BY 4.0 license. Please check copyright information on all replacement figures and update the figure caption with source information. If applicable, please specify in the figure caption text when a figure is similar but not identical to the original image and is therefore for illustrative purposes only.The following resources for replacing copyrighted map figures may be helpful: USGS National Map Viewer (public domain): http://viewer.nationalmap.gov/viewer/The Gateway to Astronaut Photography of Earth (public domain): http://eol.jsc.nasa.gov/sseop/clickmap/Maps at the CIA (public domain): https://www.cia.gov/library/publications/the-world-factbook/index.html and https://www.cia.gov/library/publications/cia-maps-publications/index.htmlNASA Earth Observatory (public domain): http://earthobservatory.nasa.gov/Landsat:
http://landsat.visibleearth.nasa.gov/USGS EROS (Earth Resources Observatory and Science (EROS) Center) (public domain): http://eros.usgs.gov/#Natural Earth (public domain): http://www.naturalearthdata.com/ 4. Please include captions for your Supporting Information files at the end of your manuscript, and update any in-text citations to match accordingly. Please see our Supporting Information guidelines for more information: http://journals.plos.org/plosone/s/supporting-information.

**Additional Editor Comments:**

Please find the Reviewers' comments below.

Reviewers' comments:

Reviewer's Responses to Questions

**Comments to the Author**

1. Is the manuscript technically sound, and do the data support the conclusions?

Reviewer #1: Yes

Reviewer #2: Yes

2. Has the statistical analysis been performed appropriately and rigorously? 

Reviewer #1: N/A

Reviewer #2: N/A

3. Have the authors made all data underlying the findings in their manuscript fully available?

Reviewer #1: Yes

Reviewer #2: Yes

4. Is the manuscript presented in an intelligible fashion and written in standard English?

Reviewer #1: Yes

Reviewer #2: Yes

5. Review Comments to the Author

**Reviewer #1:**  Line 492- 493: Will the figures be expanded in the final print? The legends in the current format are not legible

Line 501: It was nice that the author used the phrase ‘synthesize current evidence’ as another study conducted by Moyo-Chilufya et al. 2023 considered NCD and mental disorders among PLHIV in Africa (ref: https://pmc.ncbi.nlm.nih.gov/articles/PMC10570719/).

Line 574: ‘As the population ages and the prevalence of NCDs rises’, kindly provide a reference for the statement.

**Reviewer #2:**  In this scoping review, the authors explored existing evidence on the relationship between HIV, mental health issues, and non-communicable co-morbidities such as hypertension and diabetes. The review is interesting and well-written. This is not usually the case for me as a reviewer, but I have almost no substantive comments for the authors. The review is contextualised appropriately, the methods are described adequately, the results seem reliable and interesting, and the discussion is reasonable without being overly ambitious. Well done!

My only suggestion would be to include a discussion of the lack of children or adolescents living with HIV in these studies. All the studies included in the review enrolled people above 18 years of age, and many were much older. Children and young people living with HIV tend to be neglected in research, especially mental health research. For a starting point for discussion of this, see the section on “insights in absence” in DOI: 10.1093/braincomms/fcad231 which also found that children and young people were absent in research on depression/inflammation. I appreciate that many of the non-communicable diseases that were of interest to the authors are more prevalent in older adults – though this is not always the case (e.g. Type 1 diabetes or chronic respiratory issues can be of substantial impact in young people), so it’s still important to highlight the lack of evidence in children with HIV.

6. PLOS authors have the option to publish the peer review history of their article (what does this mean? ). If published, this will include your full peer review and any attached files.

**Do you want your identity to be public for this peer review?** For information about this choice, including consent withdrawal, please see our Privacy Policy .

Reviewer #1: No

Reviewer #2: No

---

## [Author Response · Author response to Decision Letter 1]

12 Jun 2025

We would like to express our sincere gratitude to the reviewers and the editorial team for their careful and thorough review of our manuscript. We greatly appreciate the time, effort, and expertise that went into evaluating our work. In the sections below, we provide detailed, point-by-point responses to each of the editorial and reviewer comments. We have carefully considered all feedback and have revised the manuscript accordingly to address the concerns and suggestions raised.

We are especially thankful for the opportunity to revise and resubmit our manuscript. The constructive critiques we received have been instrumental in enhancing the clarity, rigor, and overall quality of the paper. We believe that the manuscript is now substantially improved and hope that the revisions meet your expectations.

Comments from the Editor

Editor Comment 1

1. Is there any target population for this review-any specific age, gender, etc.? Please clarify. It would be nice to add the names of African Countries to the inclusion criteria.

Author Response to Comment 1: Thank you for your thoughtful suggestions. In designing our selection criteria, we intentionally did not impose any specific age or gender restrictions during the selection process, as our aim was to ensure the review was as inclusive and representative of the current literature as possible. By including studies across all age groups and genders, we aimed to ensure that our analysis reflects the diverse and multifaceted nature of syndemic interactions. We specifically sought to examine contributing factors relevant to both male and female individuals to reflect the broader landscape of syndemic interactions.

While all included studies enrolled participants aged 18 and older—with the exception of Carpenter et al.—we did not apply an upper or lower age limit in our inclusion criteria. As noted in Section 2.1 (Inclusion/Exclusion Criteria), we have incorporated a note indicating no restrictions were placed on age or gender. Interestingly, the absence of studies involving children and adolescents highlights a clear gap in the current literature. We have now addressed this point explicitly in the lack of syndemic literature section, emphasizing the need for future research focused on younger populations, who appear to be underrepresented in syndemics research despite the potential relevance of several non-communicable conditions in this group.

Regarding your second point, given the large number of countries represented in the included studies (approximately 46), we opted to avoid listing them all in the main text to reduce visual clutter. Instead, we have provided the full list of countries in S1 Appendix SS for reference.

Editor Comment 2

2. In Figure 2, please recheck the calculation—the studies screened (2913) – studies excluded (2706) = 207. The total shown in the figure is 188.

Author Response to Comment 2: Thank you for carefully reviewing the figure and bringing this discrepancy to our attention. We have rechecked the calculations and identified the source of the error. The figure has been revised accordingly to accurately reflect the correct number of studies. The updated version is now included in the revised manuscript.

Editor Comment 3

3. Scoping reviews do not typically assess the quality of the studies. Usually, systematic reviews appraise studies as one of the core components of their methodology. Figure 6 shows the quality assessment of 13 studies. How about other studies’ findings (for quality)?

Author Response to Comment 3: Thank you for the observation. You are correct that formal quality or risk of bias (RoB) assessments are not typically required for scoping reviews, as the primary goal is to map the breadth of evidence rather than evaluate the strength of individual findings. In our case, we conducted a targeted quality assessment of the 13 studies most directly aligned (i.e., qualitative, mixed methods, and cross-sectional) with our objective to provide insight into the methodological rigor of this subset. While we did review all included studies qualitatively for relevance and basic methodological soundness, we did not apply a formal RoB tool to the remaining studies (4), consistent with scoping review methodology. We have clarified this in the manuscript in section 3.6 Quality assessment and specified the quality assessment presented in Figure 6.

Comments from Reviewers

Reviewer 1 Comment 1

1. Line 492-493: Will the figures be expanded in the final print? The legends in the current format are not legible.

Author Response to Comment 1: Thank you for this helpful observation. We agree that clarity is essential for effective communication of the data. In response, we have revised the figure by enlarging the legend to improve legibility and ensure that all elements are clearly visible. The updated figure has been included in the revised manuscript.

Reviewer 1 Comment 2

2. Line 501: it was nice that the author used the phrase ‘synthesize current evidence’ as another study conducted by Moyo-Chilufya et al. 2023 considered NCD and mental disorders among PLHIV in Africa.

Author Response to Comment 2: Thank you for your kind words and for highlighting the relevance of the phrase in connection with the work by Moyo-Chilufya et al. (2023). We appreciate your recognition and are encouraged by the alignment of our approach with existing research in this important area.

Reviewer 1 Comment 3

3. Line 574: ‘As the population ages and the prevalence of NCDs rises’, kindly provide a reference for this statement.

Author Response to Comment 3: Thank you for catching this important detail. We have now added appropriate references to support this statement, drawing from recent literature that highlights the global trends in aging populations and the increasing burden of NCDs. These citations have been included in the revised manuscript to strengthen the contextual foundation of this point.

Reviewer 2 Comment 1

1. In this scoping review, the authors explored existing evidence on the relationship between HIV, mental health issues, and non-communicable co-morbidities such as hypertension and diabetes, The review is interesting and well-written. This not usually the case for me as a reviewer, but I have almost not substantive comments for the authors. The review is contextualized appropriately, the methods are described adequately, the results seem reliable and interesting, and the discussion is reasonable without being overly ambitious. Well done!

My only suggestion would be to include a discussion of the lack of children or adolescents living with HIV in these studies. All the studies included in the review enrolled people above 18 years of age, and many were much older. Children and young people living with HIV tend to be neglected in research, especially mental health research. For a starting point for discussion of this, see the section on “insights in absence” in DOI:10.1093/braincomms/fcad231 which also found that children and young people were absent in research on depression/inflammation. I appreciate that many of the non-communicable diseases that were of interest to the authors are more prevalent in older adults – though this is not always the case (e.g. Type I diabetes and chronic respiratory issues can be of substantial impact in young people), so it’s still important to highlight the lack of evidence in children with HIV.

Author Response to Comment 1: Thank you very much for your generous feedback and for highlighting an important gap in the literature. We greatly appreciate your thoughtful suggestion regarding the exclusion of children and adolescents living with HIV in the studies reviewed. In response we have expanded Section 4.1, to explicitly address this issue by discussing the lack of syndemic research involving younger populations living with HIV in Africa. We have also cited the article you recommended (DOI:10.1093/braincomms/fcad231) to underscore the broader concern of underrepresentation of children and adolescents in mental health and syndemic research. We agree that this is a critical area for future investigation, particularly given the unique challenges faced by younger individuals with chronic conditions such as HIV and NCDs.

---

## [Editor Report · Decision Letter 1]

20 Jun 2025

PONE-D-25-11960R1Syndemic Interactions between HIV/AIDS, Mental Health Conditions, and Non-Communicable Diseases in sub-Saharan Africa: A Scoping Review of Contributing FactorsPLOS ONE

Dear Dr. Karbasi,

Thank you for submitting your revised manuscript to PLOS ONE. We appreciate the time and effort you have put into addressing the reviewers' comments. After reviewing the updated submission, we noted that while many of the comments have been addressed thoughtfully, there are still a few points that require further attention before we can proceed. Therefore, we invite you to submit a revised version of the manuscript that addresses the points raised during the review process.

**Kindly review Figure 2 (the PRISMA flow diagram) carefully to ensure that the number of studies is accurately represented throughout the chart.**

We look forward to receiving your revised manuscript.

Kind regards,

Saima Hirani, PhD

Academic Editor

PLOS ONE
---

## [Author Response · Author response to Decision Letter 2]

23 Jun 2025

We have corrected PRISMA Flow Diagram to account for the correct number of studies and appreciate the editor for bringing this to our attention.

---

## [Editor Report · Decision Letter 2]

3 Jul 2025

Syndemic Interactions between HIV/AIDS, Mental Health Conditions, and Non-Communicable Diseases in sub-Saharan Africa: A Scoping Review of Contributing Factors

PONE-D-25-11960R2

Dear Dr. Karbasi,

We’re pleased to inform you that your manuscript has been judged scientifically suitable for publication and will be formally accepted for publication once it meets all outstanding technical requirements.

Kind regards,

Saima Hirani, PhD

Academic Editor

PLOS ONE

---

## [Editor Report · Acceptance letter]

PONE-D-25-11960R2

PLOS ONE

Dear Dr. Karbasi,

I'm pleased to inform you that your manuscript has been deemed suitable for publication in PLOS ONE. Congratulations! Your manuscript is now being handed over to our production team.

Kind regards,

on behalf of

Dr. Saima Hirani

Academic Editor

PLOS ONE